# The Impact of Glucagon-like Peptide-1 Receptor Agonists on Erectile Function: Friend or Foe?

**DOI:** 10.3390/biom15091284

**Published:** 2025-09-05

**Authors:** Dimitris Kounatidis, Natalia G. Vallianou, Eleni Rebelos, Kalliopi Vallianou, Evanthia Diakoumopoulou, Konstantinos Makrilakis, Nikolaos Tentolouris

**Affiliations:** 1Diabetes Center, First Department of Propaedeutic Internal Medicine, Medical School, National and Kapodistrian University of Athens, Laiko General Hospital, 11527 Athens, Greece; elenirebelos@gmail.com (E.R.); evitadiak@gmail.com (E.D.); kmakrila@med.uoa.gr (K.M.); ntentol@med.uoa.gr (N.T.); 2First Department of Internal Medicine, Sismanogleio General Hospital, 15126 Athens, Greece; natalia.vallianou@gmail.com; 3Turku PET Centre, University of Turku, 20014 Turku, Finland; 4Department of Clinical and Experimental Medicine, University of Pisa, 56126 Pisa, Italy; 5Clinic of Nephrology and Renal Transplantation, Medical School, National and Kapodistrian University of Athens, Laiko General Hospital, 11527 Athens, Greece; kallia_harry@hotmail.com

**Keywords:** diabetes, dulaglutide, erectile dysfunction, GIP, GLP-1 receptor agonists, hypogonadism, liraglutide, semaglutide, testosterone, tirzepatide

## Abstract

Erectile dysfunction (ED) is a common yet frequently underrecognized microvascular complication of diabetes, affecting up to three out of four individuals. Key contributing factors include advancing age, long-standing disease duration, and suboptimal glycemic control, as well as insulin resistance and androgen deficiency—the latter being particularly common in men with type 2 diabetes (T2D) and obesity. While numerous studies have investigated the effects of various antidiabetic therapies on diabetes-related ED, the results remain inconsistent, limiting definitive conclusions. In recent years, increasing attention has focused on a novel class of antidiabetic medications, namely glucagon-like peptide-1 receptor agonists (GLP-1 RAs). These agents have become central to the treatment of T2D due to their potent glucose-lowering properties and well-documented benefits on cardiovascular outcomes, and weight loss. Given these pleiotropic effects, GLP-1 RAs have been presumed to positively influence erectile function—a hypothesis supported by a growing body of experimental and clinical research. However, preliminary reports have also raised concerns about a possible association between GLP-1 RA use and ED. This narrative review aims to synthesize current evidence regarding the impact of GLP-1 RAs on erectile function, providing a platform for future research in this evolving field.

## 1. Introduction

Erectile function is a cornerstone of male sexual health, and its impairment can negatively affect overall quality of life (QoL), emotional intimacy, and relationship satisfaction. Erectile dysfunction (ED) is defined as the persistent inability to achieve or maintain an erection sufficient for satisfactory sexual performance and can significantly diminish sexual well-being [1,2]. The diagnosis of ED relies on a comprehensive clinical approach, including detailed medical and sexual history, physical examination, laboratory evaluations, and standardized assessment questionnaires [1,3]. Among the most widely used instruments is the 15-item International Index of Erectile Function (IIEF), while its abridged 5-item version (IIEF-5) provides a practical alternative for clinical settings. In addition to facilitating diagnosis, the IIEF-5 enables classification of ED severity, with scores ranging from severe ED (5–7) to no ED (22–25) [4]. First-line treatment for ED typically involves oral phosphodiesterase type 5 (PDE5) inhibitors, such as sildenafil, tadalafil, vardenafil, and avanafil. These agents act by inhibiting the PDE5 enzyme, thereby increasing cyclic guanosine monophosphate (cGMP) levels. Elevated cGMP promotes smooth muscle relaxation and enhances blood flow to the corpus cavernosum during sexual arousal, thereby facilitating penile erection [5].

ED is particularly prevalent among men with diabetes and has also been reported in individuals with prediabetes [6]. Hyperglycemia, insulin resistance (IR) and hypogonadism, particularly in the presence of obesity, constitutes the core pathophysiological triad underlying diabetes-induced ED [7]. Men with diabetes are approximately 3.5 times more likely to experience ED compared to their non-diabetic counterparts [8]. Although numerous epidemiological studies have sought to quantify the prevalence of diabetes-related ED, reported rates vary considerably, ranging from 35% to 75% [9]. A recent umbrella review encompassing data from 108,030 men with diabetes estimated a global pooled prevalence of 65.8% [10]. Notably, prevalence appears to be lower among men with type 1 diabetes (T1D), with reported rates up to 42.5% [11,12].

Several risk factors have been implicated in the development of diabetes-associated ED. These factors encompass advancing age, long-standing disease duration, and poor glycemic control as captured by high glycated hemoglobin (HbA1c) levels, and glycemic variability detected via continuous glucose monitoring (CGM). Additional risk factors include cardiometabolic comorbidities, including hypercholesterolemia, hypertension, cardiovascular disease (CVD), and obesity, as well as diabetes-related microvascular complications [11,12,13,14]. Notably, a recent study introduced a predictive model with strong performance (concordance index [C-index] = 0.827, 95% confidence interval [CI] = 0.772–0.882) for early detection of ED in subjects with type 2 diabetes (T2D), incorporating variables such as diabetes duration, carotid intima-media thickness (CIMT), and low-density lipoprotein cholesterol (LDL-C) levels [15].

Research has also explored the impact of various antidiabetic therapies on erectile function; however, the current evidence remains limited and ambiguous. In this context, increasing attention has been directed toward the potential effect of newer incretin-based therapies on erectile health, particularly glucagon-like peptide-1 receptor agonists (GLP-1 RAs) [16]. GLP-1 RAs have transformed the landscape of diabetes care by providing not only effective glycemic control but also substantial cardiovascular (CV) protection and significant weight loss [17,18]. Given that endothelial integrity and body weight are critical determinants of erectile function, it is plausible to hypothesize that these therapeutic advantages might extend to this domain as well. But does the available evidence truly support this assumption?

This narrative review examines the current evidence regarding the influence of GLP-1 RAs on erectile function. It begins with a discussion of the physiology of penile erection, and the pathophysiological mechanisms underlying diabetes-related ED, followed by a summary of the pharmacological and clinical characteristics of GLP-1 RAs. Subsequently, it presents the available preclinical and clinical data regarding their role in modulating erectile physiology. Finally, the review considers the impact of other antidiabetic drug classes on erectile function, offering a comparative perspective with GLP-1 RAs. Limitations of the existing literature are discussed, along with current challenges and future directions for research aimed at elucidating the therapeutic potential of GLP-1 RAs in this context.

## 2. Physiological Mechanisms Underlying Male Sexual Arousal

Sexual desire may be elicited by a variety of stimuli, including visual, auditory, tactile, or cognitive cues, which activate neural pathways that facilitate the relaxation of vascular and cavernosal smooth muscle. This relaxation allows for increased arterial inflow into the corpora cavernosa, a process governed by neurovascular mechanisms originating from hypothalamic centers and the sacral spinal cord (SSC) [19].

Central to this process is the neurotransmitter nitric oxide (NO), which is synthesized and released by two main categories of neurons. Non-adrenergic, non-cholinergic (NANC) neurons located in the pelvic plexus and pelvic nerves express neuronal nitric oxide synthase (nNOS) and release NO directly into adjacent vascular smooth muscle cells (VSMCs). In parallel, cholinergic parasympathetic neurons, whose cell bodies reside in the sacral parasympathetic nucleus (S2–S4) and project to the pelvic plexus and corpora cavernosa, release acetylcholine. Acetylcholine acts on M3 muscarinic receptors expressed on endothelial cells, activating endothelial nitric oxide synthase (eNOS) and promoting NO production [20,21]. Once produced, NO diffuses into neighboring VSMCs, where it stimulates soluble guanylyl cyclase (sGC), catalyzing the conversion of guanosine triphosphate (GTP) into cGMP. Elevated intracellular cGMP levels activate protein kinase G type 1 (PKG-1), which lowers cytosolic calcium concentrations and activates calcium-dependent potassium (KCa) channels, favoring VSMC hyperpolarization and smooth muscle relaxation [22,23].

Testosterone secreted by the testes exerts both genomic and non-genomic effects [24]. Through its interaction with nuclear androgen receptors (ARs) in endothelial and VSMCs, the hormone modulates gene expression to support vascular integrity. In addition, testosterone exerts non-genomic actions via membrane-bound androgen receptors (mARs) that function independently of classical AR activation [25]. These mechanisms collectively enhance endothelial NO production and stimulate the release of endothelium-derived hyperpolarizing factors (EDHFs), such as epoxyeicosatrienoic acids, hydrogen peroxide, and potassium ions, which further contribute to vasodilation within the erectile tissue [26].

As the corpora cavernosa fill with blood and expand, the resulting increase in intracavernosal pressure compresses the subtunical venous plexus beneath the tunica albuginea, restricting venous outflow and helping to maintain the erection. Additionally, contraction of pelvic floor muscles, such as the bulbospongiosus and ischiocavernosus further compresses the corpora cavernosa, enhancing penile rigidity and erection stability [27]. Figure 1 illustrates the main pathophysiological mechanisms driving male erection.

## 3. Pathophysiology of Erectile Dysfunction in Patients with Diabetes

### 3.1. Hyperglycemia

Poor glycemic control is a major contributing factor to the development of ED, regardless of diabetes type. Persistent hyperglycemia contributes to ED through several interconnected mechanisms, primarily driven by the generation of advanced glycation end products (AGEs) and reactive oxygen species (ROS) [28]. AGEs constitute a heterogeneous group of compounds formed via non-enzymatic reactions between reducing sugars (e.g., glucose) and proteins, lipids, or nucleic acids. Prolonged hyperglycemia accelerates this glycation process, resulting in the accumulation of AGEs both endogenously through metabolic pathways and exogenously from dietary sources. Over time, AGEs undergo further irreversible modifications that disrupt cellular and tissue function. These effects are mainly mediated through their interaction with the receptor for advanced glycation end products (RAGE) [29].

AGEs exacerbate ED by promoting oxidative stress and attenuating NO bioavailability, thereby impairing the relaxation of cavernosal smooth muscle [30]. This dysfunction is further exacerbated by the decreased expression and activity of both nNOS and eNOS, leading to impaired neurovascular regulation of penile erection [31,32,33]. Additionally, in diabetes, a state of NO resistance emerges, characterized by diminished responsiveness of cavernosal smooth muscle cells to NO signaling. This resistance is primarily driven by increased oxidative stress, and desensitization of the sGC–cGMP–PKG signaling cascade [34]. Moreover, hyperglycemia impairs maximal cavernosal smooth muscle relaxation in response to both acetylcholine and nitrergic nerve stimulation. This dysfunction is linked to enhanced activation of key signaling pathways and molecules, including Rho-associated kinase 2 (ROCK2), p38 mitogen-activated protein kinase (p38 MAPK), and arginase II [35]. ROCK2 signaling is critical in regulating vascular tone, cytoskeletal dynamics, intercellular adhesion, and endothelial barrier function [36]. Research has shown that in diabetic mice with heterozygous deficiency of ROCK2, these pathological changes are significantly reduced or even absent [35].

Beyond impairments in cavernosal relaxation, ROS accumulation and oxidative stress linked to AGE formation promote reductions in antioxidant responses, lipid peroxidation, DNA damage, and oxidative protein modifications [37,38]. Nitration of tyrosine residues to form 3-nitrotyrosine (3-NT) alters protein structure and function, disrupting enzymatic activity and intracellular signaling. These reactions, mediated by reactive nitrogen species (RNS), further reduce NO availability, ultimately worsening endothelial dysfunction and impairing vasorelaxation [39]. Interestingly, recent evidence supports that the combined accumulation of AGEs and ROS triggers various forms of programmed cell death, including apoptosis, pyroptosis, and ferroptosis, resulting in ED aggravation [40].

Low-intensity pulsed ultrasound (LIPA) may exert protective effects against AGEs-induced endothelial injury in the corpus cavernosum by promoting mitophagy, thereby preserving mitochondrial function and cellular homeostasis [41]. Furthermore, sustained high glucose levels might facilitate corpus cavernosum fibrosis via the transforming growth factor beta 1/SMAD family proteins/connective tissue growth factor (TGF-β1/Smad/CTGF) signaling pathway [42]. Epidemiological evidence indicates an association between diabetes and Peyronie’s disease, with diabetes recognized as a risk factor for its development [43].

The frequent coexistence of comorbidities in diabetes can intensify ED by amplifying the aforementioned pathophysiological processes. For instance, the coexistence of diabetes and hypertension might exhibit a synergistic negative effect on erectile function by enhancing oxidative stress, inflammation, and apoptosis [44]. Notably, ED may persist or even worsen despite appropriate antihypertensive treatment, as certain medications, especially thiazide diuretics, beta-adrenergic blockers, and aldosterone antagonists, have been linked to detrimental effects [45]. Moreover, hypercholesterolemia may further impair NO signaling by promoting the overactivation of nicotinamide adenine dinucleotide phosphate hydrogen (NADPH) oxidase subunits, leading to elevated superoxide production and eNOS uncoupling [46].

### 3.2. Insulin Resistance

IR may contribute to ED development, primarily through its noxious impact on endothelial function. By impairing endothelial NO synthesis and attenuating insulin-mediated vasodilation, IR disrupts the vascular mechanisms essential for erection [47]. In a model of insulin-resistant, obese mice, short-term pharmacological improvement of IR was sufficient to restore erectile function. This restoration was attributed to the normalization of both endothelium-dependent and nerve-mediated corpus cavernosum relaxation, along with reactivation of the cGMP signaling pathway and attenuation of cavernosal hypercontractility [48]. Clinical research further supports this association. In a prospective study involving 283 men with ED lasting at least six months, IR was identified in nearly half of the participants. Notably, IR was independently associated not only with the presence of ED but also with its severity [49]. Several studies, including analyses from large-scale datasets such as the National Health and Nutrition Examination Survey (NHANES), have identified surrogate markers related to IR that could potentially facilitate early risk assessment for ED. These indices include the homeostasis model assessment of insulin resistance (HOMA-IR), the triglyceride-glucose (TyG) index, as well as the metabolic score for insulin resistance (METS-IR). However, considerable heterogeneity among studies and variability in the diagnostic performance of these indices currently limit their clinical utility as predictive tools for ED [50,51,52].

### 3.3. Hypogonadism

Hypogonadism, defined by impaired testosterone production due to the dysfunction of the testes, hypothalamus, or pituitary gland, serves as a key metabolic disturbance linking diabetes and ED [53]. Both reduced circulating testosterone levels and diminished concentrations of its primary plasma binding protein, sex hormone-binding globulin (SHBG), have been identified as independent predictors of T2D development [54,55]. When present alongside obesity, androgen deficiency augments IR, creating a bidirectional loop that worsens both T2D and ED [56]. Although obesity is commonly associated with T2D, it is increasingly observed in T1D, giving rise to the concept of ‘’double diabetes’’ [57]. Moreover, hypertension and other comorbidities may further compound testosterone-driven impairments in erectile function [58].

Androgen deprivation disrupts vascular homeostasis by promoting vasoconstriction, VSMC apoptosis, adipocyte infiltration, and fibrosis of the tunica albuginea [59,60,61,62]. Within the corpora cavernosa, the absence of adequate androgen signaling leads to reduced expression of eNOS and PDE5 mRNA, with the severity of the functional consequences correlating with the extent of enzymatic downregulation [63]. Persistent high glucose levels have been linked to decreased testosterone levels, in part due to the downregulation of vascular endothelial growth factor (VEGF), which compromises testicular microcirculation, leading to Leydig cell dysfunction and blunted testosterone synthesis [64]. On the other hand, disrupted insulin signaling mitigates hypothalamic kisspeptin expression, perturbing the hypothalamic–pituitary–gonadal (HPG) axis via altered mammalian target of rapamycin (mTOR)–kisspeptin pathway activity and impaired gonadotropin-releasing hormone (GnRH) secretion [65,66].

The HPG axis is further suppressed in obesity, through increased aromatase activity in adipose depots, driving peripheral conversion of androgens to estrogens. Elevated estradiol activates hypothalamic estrogen receptors, triggering a negative feedback on gonadotropin release and further lowering testosterone output [67]. This androgen deficiency skews stromal progenitor cells (SPCs) toward adipogenic differentiation, causing ectopic lipid accumulation within erectile tissue and worsening structural dysfunction [68]. Interestingly, a recent transcriptomic analysis has identified aberrant expression patterns of adiponectin receptors in men with ED [69].

Collectively, the combined effects of prolonged hyperglycemia, obesity-driven hormonal imbalance, and androgen deprivation promote the development of a chronic pro-inflammatory milieu that favors progressive endothelial dysfunction [70]. Figure 2 presents key pathophysiological events and mechanisms underlying ED in individuals with diabetes.

## 4. GLP-1 Receptor Agonists: Therapeutic Impact and Safety Considerations

### 4.1. Pharmacological Profile and Clinical Benefits of GLP-1 Receptor Agonists

GLP-1 RAs constitute a novel class of antidiabetic medications that have reshaped the management of T2D. These agents function by mimicking the actions of endogenous GLP-1, binding to its receptor to enhance glucose-dependent insulin secretion from pancreatic β-cells while suppressing glucagon release from α-cells. This glucose-dependent mechanism offers the benefit of avoiding hypoglycemia, while simultaneously contributing to effective glycemic control [71]. GLP-1 RAs are classified into short-acting and long-acting agents based on their pharmacokinetic properties. Short-acting agents, such as exenatide twice-daily and lixisenatide, primarily target postprandial glucose excursions by delaying gastric emptying. In contrast, long-acting GLP-1 RAs, including exenatide long-acting release (exenatide LAR), liraglutide, dulaglutide, and semaglutide, provide more sustained reductions in fasting plasma glucose (FPG) by continuously enhancing insulin secretion and suppressing glucagon release [72]. In contemporary clinical practice, once-weekly injectable formulations, particularly semaglutide are preferred, due to their superior efficacy and ease of use [71].

Beyond glucose control, GLP-1 RAs have gained prominence for their CV benefits, especially in individuals with established CVD. These benefits are not limited to patients with T2D but also extend to those with obesity, even in the absence of T2D [73]. The landmark LEADER (Liraglutide Effect and Action in Diabetes: Evaluation of Cardiovascular Outcome Results) trial, which enrolled 9340 individuals with T2D at high CV risk, demonstrated that liraglutide significantly reduced the incidence of major adverse cardiovascular events (MACE) compared with placebo over a median follow-up of 3.8 years (Hazard Ratio [HR]: 0.87; 95% CI: 0.78–0.97; *p* = 0.01 for superiority; *p* < 0.001 for noninferiority) [74]. Similar CV benefits were observed in the SUSTAIN-6 trial for semaglutide. This study included 3297 patients with T2D at high CV risk, 83% of whom had established CVD, chronic kidney disease (CKD), or both. Once-weekly semaglutide significantly reduced the incidence of the primary composite outcome (first occurrence of CV death, nonfatal myocardial infarction, or nonfatal stroke) compared with placebo (6.6% vs. 8.9%; HR: 0.74; 95% CI: 0.58–0.95; *p* < 0.001 for noninferiority [75]. Prescription of semaglutide may also provide renal protective benefits in patients with T2D and CKD. Recent evidence indicated that its use reduced the risk of a composite kidney-specific outcome, including progression to kidney failure or a sustained ≥50% decline in estimated glomerular filtration rate (eGFR), by 21% (HR: 0.79; 95% CI: 0.66–0.94) and attenuated the annual decline in eGFR by 1.16 mL/min/1.73 m^2^ (*p* < 0.001) [76].

One of the most impactful therapeutic benefits of GLP-1 RAs is their capacity to induce substantial and sustained weight loss, particularly for liraglutide and semaglutide. Although the precise mechanisms underlying this effect are not fully elucidated, it is thought to involve central appetite suppression, enhanced satiety, and delayed gastric emptying [77]. High-dose formulations of these agents, specifically liraglutide at 3.0 mg and semaglutide at 2.4 mg, have been approved for the treatment of obesity, independent of glycemic status [78]. More recently, tirzepatide, a dual glucose-dependent insulinotropic polypeptide (GIP)/GLP-1 receptor agonist, has demonstrated superior weight-reducing effects compared with both placebo and semaglutide, particularly in the SURMOUNT trials, with these benefits sustained for up to 72 weeks [79,80,81].

GLP-1 RAs have also shown efficacy in metabolic dysfunction-associated steatotic liver disease (MASLD) and are now recommended as a first-line antidiabetic treatment for patients with metabolic dysfunction-associated steatohepatitis (MASH) and diabetes [82]. In a 24-week trial, dulaglutide administration produced a 26.4% relative decrease in liver fat content alongside significant reductions in gamma-glutamyl transpeptidase (GGT) levels [83]. Semaglutide has yielded even greater benefits, with its maximal dosage over 72 weeks increasing the rate of MASH resolution without fibrosis progression (59% vs. 17% with placebo; *p* < 0.001) [84]. The wide-ranging biological effects of GLP-1 RAs have further prompted investigation into other conditions, including Alzheimer’s disease (AD), where their impact appears favorable, although the current evidence remains preliminary [85].

### 4.2. Adverse Effects and Safety Profile of GLP-1 Receptor Agonists

The most frequently reported adverse effects of GLP-1 RAs are gastrointestinal, including nausea, vomiting, diarrhea, and constipation. These events are typically dose-dependent and occur most often during treatment initiation or dose escalation. They are attributed to mechanisms overlapping with the weight-reducing properties of GLP-1 RAs, particularly delayed gastric emptying, which is considered the predominant factor [86,87]. A modest increase in heart rate has also been observed, although current evidence does not support an increased risk of arrhythmia [86].

GLP-1 RAs have also been associated with pancreatitis, which has predominantly been linked to gallbladder disease as a secondary effect of weight loss [88]. In addition, evidence suggests that GLP-1 RAs may increase pancreatic enzyme secretion, while the expression and subsequent stimulation of GLP-1 receptors in pancreatic islet and exocrine duct cells have been correlated with overgrowth of the ductal epithelial cells, resulting in hyperplasia and chronic low-grade or acute inflammation, potentially leading to pancreatitis [89]. Nevertheless, the true incidence appears to be low, and several studies have questioned the existence of a causal relationship [90,91,92]. Earlier experimental data are consistent with this view, showing that although GLP-1 receptor activation can increase pancreatic mass and alter gene expression, it does not confer susceptibility to pancreatitis [93]. GLP-1 RAs have also been associated with an increased risk of pancreatic cancer, given the established link between pancreatitis and pancreatic malignancy. Nevertheless, recent evidence largely supports the safety profile of GLP-1 RAs. Regarding other malignancies, preclinical and clinical studies suggest potential anti-cancer properties through suppression of tumor initiation and progression. An important exception includes individuals with a personal or family history of medullary thyroid carcinoma (MTC), where GLP-1 RAs are contraindicated. This recommendation is primarily based on rodent studies demonstrating increased calcitonin secretion and C-cell hyperplasia, although only minimal GLP-1 receptor expression has been reported in human C-cells [94].

With the increasing adoption of GLP-1 RAs, less common adverse effects are receiving increased attention. Recent pharmacovigilance data have implicated GLP-1 RAs, particularly semaglutide, in the development of androgenetic alopecia [95,96]. The precise mechanisms remain insufficiently defined, with available data relying largely on speculations. The leading hypothesis posits that rapid weight loss may induce metabolic and nutrient alterations that trigger telogen effluvium [97]. Ocular safety has also emerged as a key consideration in GLP-1 RA therapy, encompassing worsening of diabetic retinopathy (DR), particularly in semaglutide users with pre-existing DR at baseline, as well as ischemic optic neuropathy (ION) [75,98,99]. Current evidence suggests that these effects are more likely attributable to rapid improvements in glycemic control rather than a direct pharmacological action of GLP-1 RAs. In contrast, other investigations have not confirmed this association, while preclinical research has suggested a potential protective role through modulation of the ROCK2 pathway within the retinal microvasculature [100,101,102,103]. Of note, a very recent in vitro study demonstrated that semaglutide exerted potent antioxidative and cytoprotective effects in retinal endothelial cells, thereby facilitating wound healing [104]. Figure 3 summarizes the main advantages and disadvantages of GLP-1 RA prescription.

The broader implications of GLP-1 RAs on diabetic microvascular disease, including ED, remain only partially understood. Both animal and human studies increasingly support the beneficial impact of GLP-1 RAs on erectile performance, primarily by improving penile vascular health and endothelial function, as well as through their weight-reducing properties [105,106]. Nevertheless, a limited number of studies have reported potential deterioration in erectile function following GLP-1 RA initiation [107]. Given these contradictory findings, the subsequent sections will shed light on the existing literature to determine whether current evidence permits any definitive clinical conclusions regarding the impact of GLP-1 RAs on erectile function.

## 5. Cardiovascular and Potential Erectile Implications of GLP-1 Receptor Agonists: Insights from Preclinical Research

### 5.1. Cardiovascular Effects of GLP-1 Receptor Agonists

A significant body of preclinical research has illuminated the beneficial impact of GLP-1 RAs on endothelial function, largely attributed to the widespread distribution of GLP-1 receptors on endothelial cells throughout different anatomical regions [108,109]. For instance, GLP-1 receptor expression has been identified in human umbilical vein endothelial cells (HUVECs), where GLP-1 appears to exert direct anti-inflammatory effects. These actions are primarily mediated via suppression of RAGE expression, through activation of the cyclic adenosine monophosphate (cAMP) signaling pathway [110]. Additional studies have indicated that GLP-1 enhances endothelial function in diabetic vasculature by reducing endothelial permeability and contractility, while also offering protection against oxidative stress and autophagic processes [111,112].

Endothelium-protective effects have been observed with both short- and long-acting GLP-1 RAs. For instance, exenatide has been shown to ameliorate endothelial dysfunction induced by hyperglycemia and hyperlipidemia through direct activation of eNOS via the AMP-activated protein kinase (AMPK) pathway [113]. Lixisenatide not only upregulates eNOS but may also promote VEGF expression, with evidence suggesting that some of its vascular effects may occur independently of GLP-1 receptor activation [114]. In the case of liraglutide, research demonstrated enhanced endothelial function via stimulation of the mammalian target of rapamycin complex 2/protein kinase B (mTORC2)/Akt pathway, resulting in increased NO production, elevated telomerase activity, and anti-apoptotic effects [115]. Liraglutide has also been reported to attenuate VSMC proliferation and atherosclerotic progression through AMPK activation and cell cycle arrest mechanisms [116]. Interestingly, in endothelial cells derived from individuals with diabetes, liraglutide alleviated endoplasmic reticulum stress, reduced c-Jun N-terminal kinase (JNK) activity, and restored insulin-mediated eNOS activation [117].

Once-weekly formulations have also produced favorable outcomes. Dulaglutide has been shown to reduce hyperglycemia-induced endothelial injury, potentially by preserving sirtuin 1 (SIRT1) levels and curbing mitochondrial fission [118], as well as via the downregulation of the NLR family pyrin domain containing 3 (NLRP3) inflammasome [119]. On the other hand, semaglutide has exhibited vascular protective effects by mitigating inflammatory damage to endothelial progenitor cells (EPCs), likely through suppression of microRNA-155 (MiR-155) in macrophage-derived exosomes [120]. It may also contribute to the maintenance of endothelial barrier integrity by limiting extracellular matrix components associated with heightened permeability [121]. Further findings revealed that semaglutide reversed cerebral microvascular rarefaction induced by high-fat diet and supported the structural coherence of the neurovascular unit [122]. Notably, the drug appears to outperform dietary interventions in improving endothelial and cardiac function in obesity-associated heart failure, possibly via modulation of inflammatory pathways, immune cell responses, and adipose tissue remodeling [123].

GLP-1 RAs may also confer indirect benefits on endothelial function through their weight-reducing properties. Weight loss not only improves glycemic control but also reduces ectopic adipose tissue depots, such as perivascular and epicardial fat, that exert direct deleterious effects on the vascular endothelium [124]. Weight loss also contributes to improved insulin sensitivity. Beyond these indirect effects, GLP-1 RAs have been shown to exert direct actions on insulin signaling, including enhanced glucose uptake, reduced oxidative stress, and improved lipid metabolism [125]. These mechanisms collectively underscore the central role of GLP-1 RAs in promoting CV health—mechanisms that may likewise translate to improved erectile function, given the shared pathophysiology between CVD and ED [126]. However, dedicated studies specifically evaluating the effects of GLP-1 RAs on erectile function remain scarce.

### 5.2. Potential Impact of GLP-1 Receptor Agonists on Erectile Function

Existing data point to a positive role of exenatide in facilitating NO–dependent smooth muscle relaxation within the corpus cavernosum, likely mediated by reduced NADPH oxidase activity. In a rat model of cavernosal dysfunction, Dalaklioglu et al. reported that exenatide significantly improved erectile function, correlating with restoration of endothelium-dependent and neurogenic relaxation, diminished oxidative stress and apoptosis, and normalization of signaling within the eNOS and ROCK2 pathways [127]. Similar effects have been reported for liraglutide, which targets overlapping molecular pathways involved in smooth muscle tone regulation. Beyond its vasodilatory properties, liraglutide has been shown to promote structural remodeling of penile tissue by enhancing autophagic activity, thereby facilitating tissue regeneration independently of its glucose-lowering or weight-reducing effects [128]. Furthermore, liraglutide may enhance the Akt/eNOS signaling pathway by promoting Akt-dependent phosphorylation of eNOS, a crucial process for maintaining NO bioavailability and supporting endothelial function in penile erection [129,130]. Indirect mechanisms have also been proposed, particularly in the context of androgen deficiency. In an orchiectomized rat model, liraglutide was found to improve metabolic disturbances, such as increased adiposity and impaired glucose regulation [131]. By contrast, to the best of our knowledge, there is currently no direct experimental evidence exploring the effects of dulaglutide or semaglutide on erectile function.

Emerging evidence indicates that GLP-1 RAs may have therapeutic potential in diabetes-related ED through a mitochondria-targeted, piezoelectric nanosystem. In a diabetic mouse model of ED, ultrasound-activated barium titanate (BaTiO_3_)-based nanosystem (BaTCG) generated localized piezoelectric currents that consumed protons (H^+^) on the mitochondrial outer membrane. This process reduced mitochondrial matrix H^+^ availability, collapsed the mitochondrial membrane potential, and triggered mitophagy, thereby attenuating oxidative stress, inflammation, and apoptosis. Simultaneously, the nanosystem enabled sustained release of long-acting GLP-1 RAs, which activated pancreatic β-cell receptors to enhance insulin secretion and improve glycemic control. The combined effects of restored metabolic balance, reduced mitochondrial injury, and suppressed inflammation promoted tissue regeneration within the corpus cavernosum, ultimately improving erectile function. Figure 4 provides a schematic representation of this nanosystem, highlighting the role of GLP-1 RA-mediated glycemic regulation in mitochondrial recovery and erectile tissue repair [132]. Several similar nanoplatforms have demonstrated promise in optimizing antidiabetic therapy; however, their clinical application remains investigational [133].

## 6. GLP-1 Receptor Agonists and Erectile Function: Insights from Clinical Research

### 6.1. Evidence Supporting a Beneficial Effect of GLP-1 Receptor Agonists on Erectile Function

While high-quality clinical data remain limited, accumulating evidence increasingly supports a potential therapeutic role for GLP-1 RAs in the management of ED. A small prospective, randomized, open-label study involving men with obesity-related functional hypogonadism unresponsive to lifestyle modification examined the effects of liraglutide (3 mg daily) and testosterone replacement therapy (TRT). Both interventions led to significant increases in total testosterone (TT) (+2.6 ± 3.5 nmol/L with liraglutide vs. +5.9 ± 7.2 nmol/L with TRT), with comparable improvements in sexual function, including libido and erectile performance. Furthermore, 16 weeks of liraglutide treatment outperformed TRT in promoting weight loss (−7.9 ± 3.8 kg vs. −0.9 ± 4.5 kg with TRT, *p* < 0.001), while a marked differential effect on the hypothalamic–pituitary–testicular (HPT) axis emerged, as TRT induced suppression of gonadotropins, whereas liraglutide produced a significant increase (*p* < 0.001) [134]. On the other hand, a subsequent prospective comparative study involving young men (aged 18–35) with metabolic hypogonadism, randomized to gonadotropins (Group A), liraglutide (Group B), or TRT (Group C), demonstrated the superiority of liraglutide in improving sexual function. After 4 months, Group B demonstrated significantly greater improvements in erectile function, with IIEF-5 rising from 4 ± 2 to 21 ± 4 (*p* < 0.05), while PDE5 inhibitor use declined most notably in Group B (Group A: 48%, Group B: 31%, Group C: 45%). Liraglutide treatment was associated with improved sperm quality, accompanied by a significant increase in TT of 192.9% (1.4 ± 0.6 vs. 4.1 ± 0.5 ng/mL, *p* < 0.05) and an elevation in SHBG of 157.1% (14.0 ± 3.0 vs. 36.0 ± 4.0 nmol/L, *p* < 0.05), both relative to baseline and in comparison with the other two groups at the end of treatment. In addition, patients receiving liraglutide exhibited significantly higher gonadotropin concentrations compared with the other groups, with FSH (Group A: 0.9 ± 0.2 IU/L vs. Group B: 2.6 ± 0.2 IU/L vs. Group C: 0.2 ± 0.1 IU/L, *p* < 0.05) and LH (Group A: 1.0 ± 0.3 IU/L vs. Group B: 3.2 ± 0.2 IU/L vs. Group C: 0.3 ± 0.1 IU/L, *p* < 0.05) [135].

The therapeutic benefits of GLP-1 RAs have also been reported when used as an adjunct to metformin. In a retrospective one-year cohort study involving male outpatients with T2D, Lisco et al. found that the combination of GLP-1 RAs with metformin, compared with metformin monotherapy, produced marked increases in TT (+41.41 ± 6.11 ng/dL, *p* < 0.0001) and free testosterone (fT) (+0.44 ± 0.09 ng/dL, *p* < 0.0001) levels. These hormonal changes were accompanied by a significant rise in IIEF-5 score (+2.26 ± 0.26, *p* < 0.0001). Interestingly, the magnitude of benefit was greater in men with higher baseline IIEF-5 score (*p* = 0.045), those presenting with carotid artery stenosis (*p* = 0.045), and those who experienced weight loss during treatment (*p* = 0.013) [106]. These results corroborate an earlier observational study by the same research group, which further demonstrated that the addition of liraglutide or dulaglutide to metformin enhanced erectile function, irrespective of baseline gonadal status (*p* < 0.01) [136].

Liraglutide also appears to provide additional benefits when used alongside TRT and metformin in men with T2D, obesity, hypogonadism, and recent-onset ED. After one year of combined TRT and metformin, patients experienced significant improvements in erectile function (*p* < 0.01), although none reached the glycemic target (HbA1c < 7.5%). Among the 43 participants, 26 showed a limited glycemic response and were considered poor responders. These individuals received liraglutide at 1.2 mg daily during the second year, while the remaining 17 continued on TRT and metformin combination. At the end of the second year, patients who remained on TRT and metformin exhibited no further gains in IIEF scores and experienced significant increases in HbA1c (*p* < 0.05) and body weight (*p* < 0.04). In contrast, subjects receiving liraglutide not only reached glycemic targets and lost weight (*p* < 0.01) but also showed additional increases in SHBG (*p* < 0.05) and TT (*p* < 0.01), with concurrent marked improvements in erectile function [137]. Genetic data provide further support for a causal link between GLP-1 RA use and reduced ED risk. A Mendelian randomization study found genetically proxied GLP-1 RA exposure to be inversely associated with ED (OR: 0.493; 95% CI: 0.430–0.565; *p* < 0.001), with mediation analysis indicating that conventional risk factors such as T2D, obesity, hypertension, and CVD explained only a small portion of the observed effect [138].

To date, the main randomized control trial (RCT) addressing erectile outcomes with GLP-1 RA therapy is an exploratory analysis of the REWIND (Researching Cardiovascular Events With a Weekly INcretin in Diabetes) trial. Among 5312 men with T2D, 3725 completed baseline and follow-up IIEF assessments. Dulaglutide treatment was associated with a modest but significant reduction in moderate-to-severe ED compared with placebo (21.3 vs. 22.0 per 100 person-years; HR: 0.92, 95% CI: 0.85–0.99; *p* = 0.021). Dulaglutide also attenuated erectile function decline over time, with a least squares (LS) mean difference in erectile function subscore of 0.61 (95% CI: 0.18–1.05; *p* = 0.006) [139].

### 6.2. Evidence Suggesting a Potential Adverse Impact of GLP-1 Receptor Agonists on Erectile Function

Despite increasing evidence supporting the beneficial effects of GLP-1 RAs on erectile function, a limited number of studies have raised concerns regarding potential adverse sexual outcomes. A retrospective cohort study using the TriNetX Research database evaluated the incidence of ED in 3094 non-diabetic men with obesity (aged 18–50) prescribed semaglutide for weight management, matched to an equal number of semaglutide-naïve men with obesity. After excluding patients with prior ED, T2D, or confounding pharmacologic treatments, semaglutide use was associated with a higher incidence of newly diagnosed ED or initiation of PDE5 inhibitor therapy (1.47% vs. 0.32%; RR: 4.5). Reported rates of testosterone deficiency were also higher among semaglutide users (1.53% vs. 0.80%; RR: 1.9). While ED and testosterone deficiency were uncommon in younger men, non-diabetic men with obesity on semaglutide showed a higher risk. Thus, the authors emphasized the importance of informing patients, particularly those at elevated risk, about this potential adverse effect, while also highlighting the need for further research to elucidate underlying mechanisms and optimize therapeutic benefits [140].

Additional signals have emerged from a cross-sectional analysis of the FDA Adverse Event Reporting System (FAERS) database (2003–2024). This pharmacovigilance study detected 182 GLP-1 RA-related sexual adverse events, extending beyond ED, to include reduced libido and orgasmic disorders. Most reports involved men aged 40–60 with T2D being the main indication of use (43.9%). Exenatide (24.2%) and semaglutide (21.4%) were the most frequently implicated agents, followed by liraglutide (20.9%) and dulaglutide (18.7%). Among the different types of sexual dysfunction identified, ED accounted for the majority, representing 71.4% of cases. Although the analysis revealed a statistically significant association (*p* < 0.0001), the study had important limitations, including those inherent to the FAERS database. Furthermore, the signal strength was weak, insufficient to indicate a significant clinical risk, highlighting the need for cautious interpretation of the study results [107]. Table 1 summarizes the main characteristics and findings of clinical studies evaluating the impact of GLP-1 RAs on erectile function.

## 7. The Influence of Other Antidiabetic Drugs on Erectile Function: Comparative Evidence with GLP-1 Receptor Agonists

### 7.1. Erectile Effects of Other Antidiabetic Therapies: Mechanistic Insights and Clinical Evidence

Multiple studies have examined the effects of antidiabetic therapies on erectile function, but findings remain inconsistent and at times contradictory. Metformin, for example, has demonstrated both beneficial and adverse effects. In a preclinical model of angiotensin II–induced ED, metformin normalized corpus cavernosum contractility and enhanced smooth muscle relaxation through increased eNOS phosphorylation [141]. When combined with antioxidants, it improved intracavernosal pressure and eNOS activity, while reducing mitochondrial autophagy, AGE generation, and the collagen-to-smooth muscle ratio in erectile tissue [142]. Conversely, short-term metformin use has been linked to reduced testosterone and libido, potentially blunting the testosterone rise typically associated with improved glycemic control [143,144].

Pioglitazone, an insulin sensitizer that functions as a peroxisome proliferator-activated receptor gamma (PPAR-γ) agonist, may exert protective effects against ED [145,146]. PPAR-γ is expressed in VSMCs, where its activation reduces oxidative stress and suppresses pro-inflammatory signaling [147]. Pioglitazone also attenuated endothelin-1–induced vasoconstriction by downregulating cyclooxygenase-2 (COX-2) via activator protein-1 (AP-1) and nuclear factor kappa-light-chain-enhancer of activated B cells (NF-κB), while enhancing NO bioavailability [148]. In a rat model of bilateral cavernous nerve injury, it restored erectile function via insulin-like growth factor-1 (IGF-1) signaling activation [149]. Clinically, pioglitazone has been shown to improve erectile response in men with sildenafil-resistant ED [150].

Insulin therapy, particularly when combined with antioxidants, has been shown to mitigate ED by inhibiting the AGEs–RAGE–oxidative stress axis [151]. It may also promote recovery by modulating apoptotic protein expression and by restoring androgen receptor expression in penile tissue [152,153]. Evidence on insulin secretagogues such as SUs indicates mixed effects. Genetic studies suggest an increased ED risk with gliclazide [154], possibly due to SU-mediated inhibition of ATP-sensitive potassium channels essential for prostaglandin E1 (PGE1)- and cAMP-dependent vasodilation [155]. Conversely, SUs have been associated with increased testosterone, enhanced libido, and improved erectile function [156]. Notably, adulterated sexual enhancement products containing glibenclamide or glyburide have caused outbreaks of severe hypoglycemia [157,158].

The role of dipeptidyl peptidase-4 inhibitors (DPP-4is) in erectile performance remains uncertain, although preliminary evidence suggests vascular benefits, including endothelial repair via endothelial progenitor cells (EPCs) mobilization [159]. Saxagliptin has been demonstrated to improve ED by upregulating stromal cell-derived factor 1 (SDF-1), which enhances EPC homing, and by activating phosphoinositide 3-kinase/protein kinase B (PI3K)/Akt signaling, thereby diminishing oxidative stress and apoptosis [160]. Additional research has revealed that toleragliptin improved erectile responses in a vincristine-induced ED model via increased nNOS expression and endothelial signaling [161]. Conversely, the VILDA study reported a 22.8% incidence of ED over two years with vildagliptin use [162].

Sodium-glucose cotransporter-2 inhibitors (SGLT-2is) are well recognized for their CV benefits, including improved arterial stiffness and endothelial function [163]; however, their effects on sexual health remain ambiguous. Dapagliflozin and empagliflozin have been shown to restore NO availability under tumor necrosis factor-alpha (TNF-α)-induced endothelial inflammation through ROS reduction. Interestingly, these effects appear independent of eNOS activity or barrier integrity [164,165]. Data on canagliflozin are limited, though one study suggested potential interaction with tadalafil metabolism via cytochrome P450 3A4 inhibition, warranting caution in coadministration [166]. Clinically, dapagliflozin improved erectile function and enhanced tadalafil responsiveness in men with T2D [167]. In contrast, a large observational study reported increased incident ED risk with SGLT-2i therapy (adjusted HR: 1.55; 95% CI: 1.40–1.72) [168].

### 7.2. Potential Comparative Advantage of GLP-1 Receptor Agonists in Erectile Function

Although limited, comparative evidence suggests that GLP-1 RAs may exert superior effects on erectile function relative to other antidiabetic agents. In a trial comparing exenatide plus metformin with glimepiride plus metformin, TT increases over 3 months were significantly greater with exenatide (121.72 ± 56.73 ng/dL vs. 34.67 ± 16.30 ng/dL) [169]. Another study reported higher IIEF-5 scores in men treated with GLP-1 RAs versus insulin (16.7 ± 4.7 vs. 12.9 ± 6.2; *p* = 0.02), a difference that remained significant after adjustment for age and diabetes duration (*p* = 0.01). No differences in erectile function were observed between metformin, SUs, DPP-4is, or SGLT-2is [170].

The comparative advantage of GLP-1 RAs has also been supported by recent systematic reviews and meta-analyses. A meta-analysis by Yang et al. demonstrated superiority of GLP-1 RAs over metformin in improving erectile function (*p* = 0.02), with greater benefits in men with BMI ≥ 30 kg/m^2^ (*p* = 0.02) [171]. Similarly, a meta-analysis by Rastrelli et al., including 4112 patients, showed that observational data supported improvements in erectile health with GLP-1 RAs, SGLT-2is, and metformin, though not superior to lifestyle interventions. In contrast, RCTs demonstrated significant IIEF improvements with metformin, pioglitazone, and dulaglutide. The authors suggested that GLP-1 RAs and SGLT-2is may be particularly effective in advanced metabolic dysfunction, whereas metformin may remain preferable for less complex cases [172]. The same research group also found that GLP-1 RA therapy significantly increased TT, fT, SHBG, and gonadotropin levels, with concurrent improvements in erectile function. Meta-regression analysis demonstrated a strong correlation between GLP-1 RA–induced weight loss and TT elevation, implicating weight reduction as a principal mediator (*p* < 0.0001) [173].

These findings are further supported by two additional systematic reviews and meta-analyses, both highlighting the favorable impact of GLP-1 RAs on functional hypogonadism, largely attributable to weight loss [174,175]. Van Cauwenberghe et al. [174] reported minimal or no hormonal effects from other antidiabetic agents. Notably, an individual study within this review yielded insights such as the neutral impact of exenatide on serum testosterone in men with T2D [176]. In contrast, Salvio et al. found that while androgenic effects of GLP-1 RAs were comparable to those of other antidiabetic therapies, they offered superior benefits in terms of BMI reduction, increased gonadotropin levels, and improved erectile function scores [175]. Table 2 presents the data, findings, and conclusions of these meta-analyses.

## 8. Research Gaps, Challenges, Future Perspectives, and Conclusions

### 8.1. Evaluating Mechanistic and Clinical Evidence Supporting GLP-1 Receptor Agonists–Mediated Erectile Benefits

ED is recognized as an early and independent predictor of CVD, with evidence indicating a 45% increased risk among individuals with ED [177]. The two conditions share overlapping pathophysiological mechanisms, including systemic inflammation, endothelial dysfunction, and atherosclerosis [178]. Current experimental data on the role of GLP-1 RAs in ED remains limited, primarily focusing on exenatide and liraglutide, with an emphasis on vascular outcomes [127,128,129,130]. On the other hand, understanding the mechanisms through which GLP-1 RAs exert their beneficial effects on CVD provides a theoretical framework for their potential effects on erectile function.

Potential benefits on erectile performance may arise from glycemic control–related mechanisms, including inhibition of AGE generation and reduction of oxidative stress [179], as well as preservation of mitochondrial integrity via modulation of programmed cell death pathways, such as pyroptosis and ferroptosis [180,181]. Furthermore, GLP-1 RAs may exert synergistic anti-inflammatory and antioxidant effects through reduced neutrophil extracellular trap (NET) formation and inhibition of NLRP3 inflammasome activation [182]. In diabetic ED models, a transcriptomic analysis demonstrated downregulation of genes involved in smooth muscle contractility and mitochondrial function, alongside upregulation of Wnt signaling and extracellular matrix remodeling [183].

Additional mechanisms may involve the modulation of key signaling pathways. The protein kinase A (PKA) pathway supports NO–mediated smooth muscle relaxation via cAMP signaling [184], whereas the Hippo–YAP1 axis regulates endothelial integrity, glucose metabolism, and fibrosis [185]. GLP-1 RA–mediated modulation of these pathways has shown benefits in other models and may potentially extent to erectile health [186,187]. Beyond endothelial dysfunction, pericyte impairment has been identified as a contributor to diabetic ED [188]. Given that GLP-1 receptors are expressed on cerebral pericytes, where they mediate vascular stabilization, similar effects in penile pericytes are plausible [189]. Moreover, the detection of GLP-1 receptors in testicular tissue raises the possibility of effects on gonadal function, mediated not only through hormonal pathways [190]. Although preclinical models frequently employ dosing regimens that limit direct translational applicability, they remain essential for delineating pharmacodynamic mechanisms, including those underlying the well-established CV benefits of GLP-1 RAs, which could also have implications for ED [191].

On the clinical front, an increasing number of studies in recent years have suggested favorable effects of GLP-1 RAs on erectile and sexual function, as well as on reproductive outcomes in both men and women [192]. However, these observations are subject to significant methodological constraints. Clinical trials investigating the effects of GLP-1 RAs on erectile function are typically limited by small sample sizes, short follow-up durations, and lack of evaluation of dose–response and exposure–response relationships. Furthermore, most available studies are observational in design and display considerable heterogeneity. To date, no RCT has been conducted with erectile function as a predefined primary endpoint [106,134,135,136,137,138,139].

These limitations also apply to comparative analyses of GLP-1 RAs versus other antidiabetic agents, for which the impact on erectile function remains uncertain. A plausible explanation for the apparent advantage of GLP-1 RAs may involve their superior weight-loss efficacy relative to other antidiabetic drugs, particularly SGLT-2is, which themselves confer substantial CV benefits. Few studies have specifically evaluated symptoms of androgen deficiency, and reported changes in testosterone levels are generally modest and subject to variability in study design [169,170,171,172,173,174,175]. Finally, the reliability and validity of instruments used to assess ED, such as the IIEF, remain debated, with evidence suggesting important limitations [193].

### 8.2. Assessing Emerging Concerns and Theoretical Risks of GLP-1 Receptor Agonists on Erectile Function

While the preponderance of available evidence suggests a beneficial effect of GLP-1 RAs on erectile function, rare reports of potential harm warrant careful consideration, even though the underlying studies face significant constraints, restricting causal inference. For instance, Able et al. relied on claims-based International Classification of Diseases, Tenth Revision (ICD-10), without verification of medication adherence. Lack of confirmation of actual drug intake limits the ability to draw conclusions about long-term effects on sexual health. Additional confounders included baseline BMI differences and the inability to distinguish between specific forms of sexual dysfunction [140]. On the other hand, although the FAERS analysis identified a statistically significant association between GLP-1 RA use and sexual dysfunction, the signal was weak, and inherent database limitations, including reporting bias and residual confounding, were present [107]. Combining the results of these two studies suggests that, while the overall risk of ED appears low or even uncertain, the increasing use of GLP-1 RAs highlights the need for ongoing clinical vigilance [107,140,194]. Of note, a recent U.S. surveillance study of emergency department visits for semaglutide-related adverse effects reported no cases of ED, though such visits are inherently uncommon [195,196].

To date, no experimental studies have demonstrated a harmful effect of GLP-1 RAs on erectile function. Consequently, the mechanisms through which GLP-1 RAs might negatively influence sexual health remain uncertain. It has been proposed that GLP-1 receptors in brain regions regulating appetite and reward may modulate neurotransmitters, such as dopamine and serotonin, potentially influencing sexual arousal [107,197]. Additional hypotheses include the rapid correction of glycemia following GLP-1 RA administration, similar to the exacerbation of DR, as well as long-term effects of prior hyperglycemia via persistent changes in endothelial gene expression, often referred to as “glucose memory” [198,199]. Other potential mechanisms involve altered testosterone pulsatility and GLP-1–mediated smooth muscle relaxation in penile tissue [140,200]. The development of ED may also be linked to the weight-lowering features of GLP-1 RAs, which can cause significant lean body mass loss, occasionally progressing to sarcopenia, a factor associated with ED [201,202]. On the other hand, in normal-weight individuals, caloric restriction has been linked to reduced testosterone levels, suggesting that GLP-1 RA use in such T2D patients may contribute to ED [203,204].

Overall, although caution is warranted, the aforementioned data should not dissuade clinicians from prescribing GLP-1 RAs, as such an association is not firmly established, and even in cases with semaglutide, the incidence was particularly low, approximately 1.4% [140,205]. In instances where ED develops following GLP-1 RA therapy, discontinuation is indicated, accompanied by a thorough urological evaluation. According to the 2025 European Association of Urology (EAU) guidelines, this evaluation should include a comprehensive medical and sexual history as well as early-morning fasting TT measurement. PDE5 inhibitors constitute the first-line treatment, while adjunctive TRT may be considered for hypogonadal non-responders, with careful attention to potential side effects, such as gastrointestinal symptoms, increased blood pressure, and thrombosis [206]. From a diabetology perspective, a reasonable approach involves substituting GLP-1 RA therapy with an alternative antidiabetic agent. If erectile function and testosterone levels improve but GLP-1 RA therapy remains clinically necessary, re-initiation at a previously tolerated lower dose or switching to a different GLP-1 RA may be considered, with close monitoring and informed consent. Notably, the timeline for testosterone recovery remains uncertain and may be prolonged [207]. Among available GLP-1 RAs, dulaglutide may represent the safest option, with even favorable effects [139], further supported by a recent RCT showing that in eugonadal normal-weight men, dulaglutide did not adversely affect sexual desire, the HPG axis, or sperm parameters compared with placebo. However, the study included a small number of participants and had a limited follow-up of only one month [208]. Despite these various hypotheses and therapeutic considerations, it is clear that well-designed, long-term RCTs are needed to provide definitive answers.

### 8.3. Comparing GLP-1 Receptor Agonists with Other Weight Loss Interventions

Assessment of the impact of GLP-1 RAs on erectile function is further limited by the lack of comparative data with other weight loss interventions. Theoretically, tirzepatide may confer additional erectile health benefits through its GIP component. However, the vascular effects of GIP remain poorly defined due to inconsistent GIP receptor (GIPR) expression across different endothelial beds [209]. Furthermore, while both GIP and GLP-1 exert protective effects on the vasculature under normal glucose levels, only GLP-1 enhances endothelial function in states of hyperglycemia. This occurs via increased eNOS expression and subsequent NO production through cAMP signaling [210]. This divergence may have implications for diabetes-associated ED, where NO bioavailability is essential. Moreover, although GIPR is expressed in the testes in preclinical models, its role in human reproductive physiology appears limited based on gene expression data [211]. On the other hand, tirzepatide may offer superior benefits over GLP-1 RAs due to its greater efficacy in weight loss. In a pilot study involving men with obesity and metabolic hypogonadism, tirzepatide improved both erectile function and hormonal profiles (*p*  <  0.05), outperforming lifestyle modification and TRT [212]. It is worth noting that pharmacovigilance data, which linked GLP-1 RA administration with ED development, reported tirzepatide in 14.3% of cases—a lower proportion compared to liraglutide, dulaglutide, and semaglutide. Nevertheless, the annual distribution of adverse event reports revealed that tirzepatide exceeded semaglutide in 2023 and early 2024, consistent with its rising clinical use [107].

Relevant insights also arise from non-pharmacological weight loss interventions, such as metabolic bariatric surgery (MBS). Evidence suggests that MBS improves erectile function through mechanisms related to weight loss and T2D remission. On the contrary, although postoperative testosterone levels increase, they do not seem to independently improve ED [213]. Notably, metabolomic studies indicate that GLP-1 RAs induce modest metabolic changes primarily related to glycemic control and appetite, whereas MBS triggers broader shifts in gut hormone signaling and nutrient sensing [214,215]. Despite emerging evidence, comparative data evaluating the effects of GLP-1 RAs, tirzepatide, and MBS on erectile function remain lacking.

In conclusion, ED represents a common yet frequently underrecognized microvascular complication of diabetes, influenced by a broad spectrum of risk factors. Emerging preclinical and clinical evidence suggests that GLP-1 RAs may exert beneficial effects on erectile function, potentially exceeding those of other antidiabetic therapies. Nevertheless, conclusions remain premature. Concurrently, recent reports have raised concerns regarding possible ED development among GLP-1 RA users. However, these observations are limited by methodological shortcomings, and the strength of the association appears weak. Further mechanistic investigations are warranted to delineate the underlying biological pathways, while rigorously designed, long-term RCTs with erectile function as a predefined primary endpoint are essential. Until such data become available, clinicians should remain vigilant for symptoms of ED following the initiation or dose titration of GLP-1 RAs and be prepared to implement appropriate therapeutic strategies. Figure 5 summarizes the key features of diabetes-related ED and outlines the current evidence regarding the effects of GLP-1 RAs on erectile function.

## Figures and Tables

**Figure 1 biomolecules-15-01284-f001:**
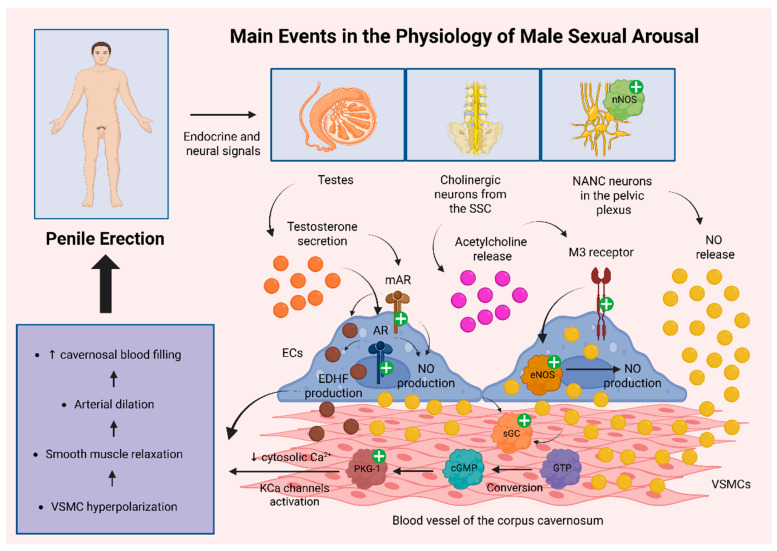
Key pathophysiological mechanisms underlying penile erection. NO plays a central role in the regulation of male erection, with two primary sources: (1) activation of nNOS in NANC neurons of the pelvic plexus, leading to direct NO release into VSMCs, and (2) activation of eNOS in endothelial cells upon stimulation of muscarinic M3 receptors by Ach released from cholinergic neurons in the SSC, with NO subsequently diffusing to VSMCs. In both cases, NO activates sGC in VSMCs, catalyzing the conversion of GTP to cGMP. Elevated cGMP levels activate PKG-1, which in turn decreases cytosolic calcium concentrations and activates KCa channels, resulting in VSMC hyperpolarization. Testosterone also enhances NO production via activation of androgen receptors (ARs and mARs), contributing to both nNOS and eNOS pathways. Additionally, testosterone stimulates the production of EDHFs, which further support VSMC hyperpolarization. The resulting hyperpolarization induces smooth muscle relaxation and vasodilation, leading to increased cavernosal blood flow and penile erection [20,21,22,23,24,25,26]. Abbreviations: Ach: acetylcholine; AR: androgen receptor; cGMP: cyclic guanosine monophosphate; ECs: endothelial cells; EDHF: endothelium-derived hyperpolarizing factor; eNOS: endothelial nitric oxide synthase; GTP: guanosine triphosphate; KCa: calcium-dependent potassium; M3 receptor: muscarinic M3 receptor; mAR: membrane-bound androgen receptor; NANC: non-adrenergic, non-cholinergic; nNOS: neuronal nitric oxide synthase; NO: nitric oxide; PKG-1: protein kinase G type 1; sGC: soluble guanylyl cyclase; SSC: sacral spinal cord; VSMC: vascular smooth muscle cell; +: activation; ↑: increase; ↓: decrease. Created in BioRender. Kounatidis, D. (2025) https://BioRender.com/rykt1qj (assessed on 27 August 2025).

**Figure 2 biomolecules-15-01284-f002:**
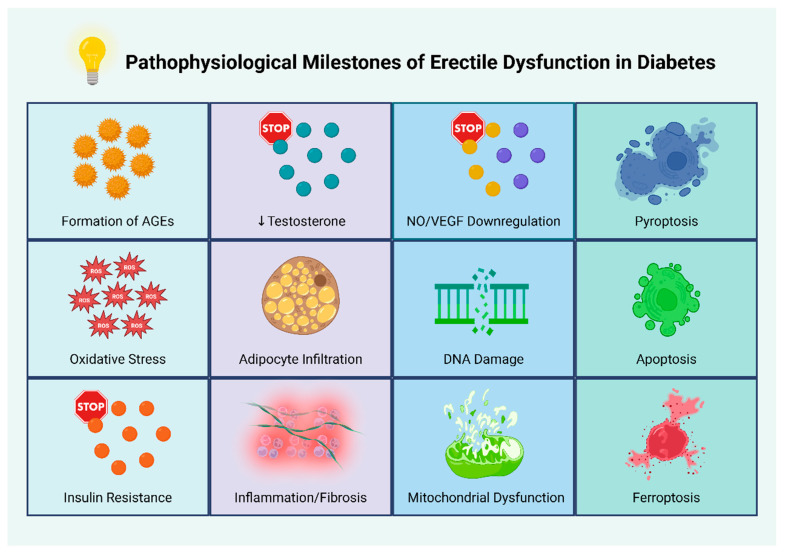
Pathophysiological milestones of erectile dysfunction in diabetes. Abbreviations: AGEs: advanced glycation end products; NO: nitric oxide; ROS: reactive oxygen species; VEGF: vascular endothelial growth factor [28,29,30,31,32,33,34,35,36,37,38,39,40,41,42,43,44,45,46,47,48,49,50,51,52,53,54,55,56,57,58,59,60,61,62,63,64,65,66,67,68,69]. Created in BioRender. Kounatidis, D. (2025) https://BioRender.com/ebz5ezk (assessed on 28 July 2025).

**Figure 3 biomolecules-15-01284-f003:**
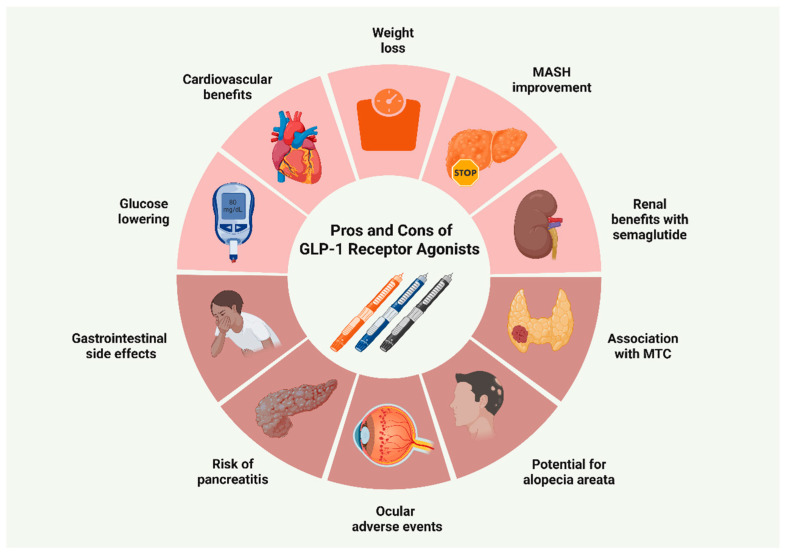
Benefits and drawbacks of GLP-1 receptor agonist use. Abbreviations: GLP-1: glucagon-like peptide-1; MASH: metabolic dysfunction-associated steatohepatitis; MTC: medullary thyroid carcinoma [71,73,76,77,82,86,90,94,95,96,98,99]. Created in BioRender. Kounatidis, D. (2025) https://BioRender.com/ez8dhoa (assessed on 28 July 2025).

**Figure 4 biomolecules-15-01284-f004:**
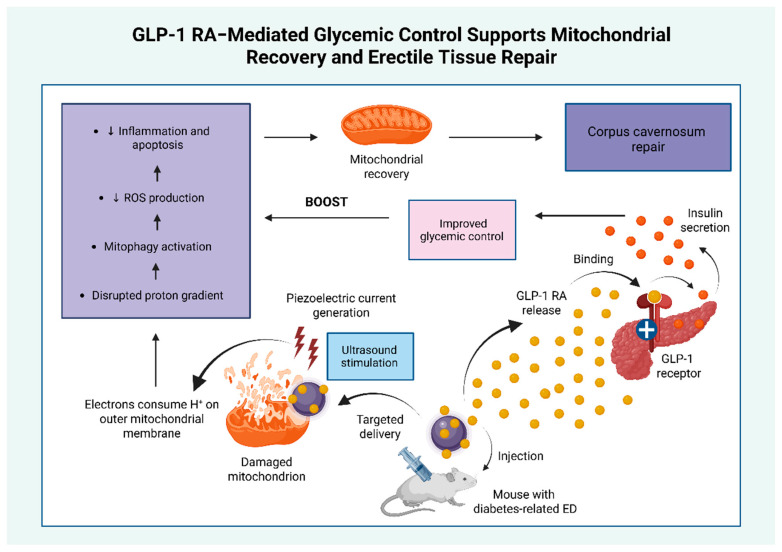
Nanotechnology-based GLP-1 RA delivery system supports mitochondrial recovery and erectile tissue repair. This figure illustrates how the improvement in glycemic control mediated by the action of GLP-1 RAs enhanced the anti-oxidant, anti-inflammatory, and anti-apoptotic effects of a mitochondria-targeted piezoelectric nanosystem, providing an additional benefit to the overall amelioration of erectile dysfunction [132]. Abbreviations: ED: erectile dysfunction; GLP-1: glucagon-like peptide 1; GLP-1 RA: glucagon-like peptide 1 receptor agonist; H^+^: protons; ROS: reactive oxygen species; +: activation. ↓: reduction. Created in BioRender. Kounatidis, D. (2025) https://BioRender.com/agueqv0 (assessed on 20 July 2025).

**Figure 5 biomolecules-15-01284-f005:**
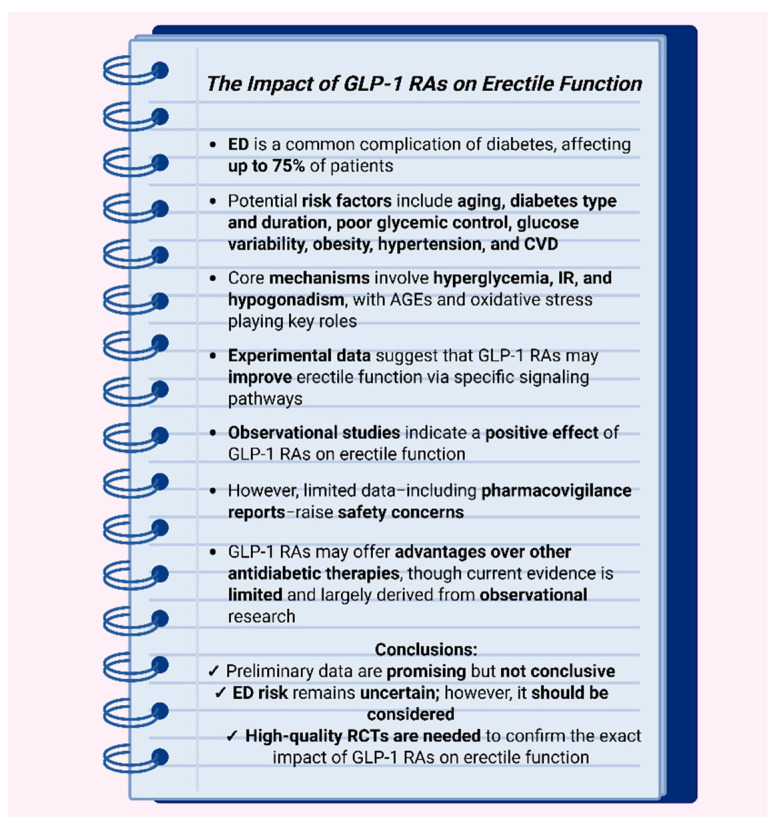
Diabetes-associated erectile dysfunction and the role of GLP-1 receptor agonists: An evidence-based snapshot. Abbreviations: AGEs: advanced glycation end products; CVD: cardiovascular disease; ED: erectile dysfunction; GLP-1 RA: glucagon-like peptide-1 receptor agonist; RCT: randomized controlled trial. Created in BioRender. Kounatidis, D. (2025) https://BioRender.com/21i8b6j (assessed on 27 August 2025).

**Table 1 biomolecules-15-01284-t001:** Main clinical studies assessing the impact of GLP-1 receptor agonists on erectile function.

Author, Year	Study Design	Results	Conclusions
Giagulli et al., 2015 [137]	✓ Retrospective observational study ✓ Objective: To evaluate whether adding liraglutide to lifestyle changes, metformin, and TRT improves erectile function ✓ Population: 43 men aged 45–59 with obesity, T2D, overt hypogonadism, and recent-onset ED ✓ Groups (post-pubertal hypogonadism *n* = 30, pre-pubertal hypogonadism *n* = 13): ▪ Year 1 (all): TRT 1 g/12 weeks plus metformin 2–3 g/day ▪ Year 2: good responders continued TRT plus metformin; poor responders added liraglutide 1.2 mg/day ✓ Duration: 2 years	✓ After 1 year: ▪ All patients showed improved erectile function and metabolic parameters ▪ No patients achieved HbA1c < 7.5% ▪ TT normalized to young adult range ✓ Good responders after 2 years: ▪ No further improvement in IIEF score ▪ ↑ HbA1c (*p* < 0.05) and ↑ body weight (*p* < 0.04) ✓ Liraglutide group after 2 years: ▪ ↓ HbA1c < 7.5% ▪ ↓ in body weight (*p* < 0.01) ▪ ↑ SHBG (*p* < 0.05) ▪ ↑ TT (*p* < 0.01) ▪ ↑ in IIEF score	✓ TRT combined with metformin initially improves erectile and metabolic parameters, regardless of hypogonadism onset ✓ In patients with poor metabolic control, adding liraglutide restores testosterone, achieves glycemic targets, reduces body weight, and enhances erectile performance
Jensterle et al., 2019 [134]	✓ Prospective, randomized, open-label study ✓ Objective: To compare the effects of liraglutide versus TRT in men with obesity-related functional hypogonadism, who were poor responders to lifestyle modification ✓ Population: 30 men with obesity (BMI 41.2 ± 8.4 kg/m^2^) and functional hypogonadism, aged 46.5 ± 10.9 years ✓ Groups: ▪ 50 mg 1% TRT gel daily ▪ 3.0 mg liraglutide daily ✓ Duration: 16 weeks	✓ ↑ TT in both groups: ▪ TRT: +5.9 ± 7.2 nmol/L ▪ Liraglutide: +2.6 ± 3.5 nmol/L ✓ Both arms improved sexual function ✓ Liraglutide group showed ↑ LH and FSH (*p* < 0.001 for between-treatment effect) ✓ Weight loss: ▪ TRT: 0.9 ± 4.5 kg ▪ Liraglutide: 7.9 ± 3.8 kg ✓ MetS resolved in 2 liraglutide patients vs. 0 in TRT	✓ When lifestyle modification fails and MBS is not indicated, liraglutide should be preferred over TRT for overall health improvement in men with obesity-related functional hypogonadism
Bajaj et al., 2021 [139]	✓ Exploratory analysis of the double-blind, randomized, placebo-controlled REWIND trial ✓ Objective: To evaluate the incidence, prevalence, and progression of ED in men with T2D receiving dulaglutide vs. placebo, and to examine whether the drug’s impact on erectile function aligns with its broader effects on diabetes-related outcomes ✓ 3725 men with T2D: ▪ Mean age 65.5 years ▪ 39.9% with established CVD ▪ 56.5% presenting with moderate to severe ED at baseline ✓ Duration: approximately 5 years	✓ Dulaglutide was associated with a lower incidence of moderate/severe ED compared with placebo (21.3 vs. 22.0 per 100 person-years; HR 0.92, 95% CI 0.85–0.99; *p* = 0.021) ✓ Dulaglutide group showed a smaller decline in IIEF scores compared with placebo (LS mean difference 0.61; 95% CI 0.18–1.05; *p* = 0.006).	✓ Long-term dulaglutide treatment may lower the risk of developing moderate or severe ED in men with T2D
Lisco et al., 2022 [136]	✓ Retrospective observational study ✓ Objective: To explore the impact of GLP-1 RAs (liraglutide and dulaglutide) on erectile function in men with T2D, both with and without baseline hypogonadism, when used as an add-on to metformin ✓ Population: 110 men with T2D and ED: ▪ Men with hypogonadism (HP): *n* = 48 and eugonadal patients (EP): *n* = 62 ▪ Mean age 51–64 years ▪ Diabetes duration 5–10 years ▪ 6% with established CVD ▪ Patients with eGFR < 60 mL/min per 1.73 m^2^ were excluded ✓ Study protocol: ▪ Men with HbA1c < 7.2% received metformin (2 g/day) ▪ Men with HbA1c > 7.2% received a GLP-1RA (52% liraglutide, 1.2 mg/day; 48% dulaglutide, 1.5 mg/week) plus metformin ✓ Groups: ▪ 28 HP were treated with GLP-1RA plus metformin (HPs) ▪ 20 HP were treated with metformin alone (HPc) ▪ 38 EP were treated with GLP-1RA plus metformin (EPs) ▪ 30 EP were treated with metformin alone (EPc) ✓ Duration: 1 year	✓ HPs and EPs: Significant ↓ in HbA1c (−0.7 ± 0.3%; *p* < 0.001) ✓ HPc and EPc: Slight ↑ in HbA1c (+0.4 ± 0.2) ✓ HPs and EPs: Significant improvement in erectile function (↑ IIEF-5 score, all *p* < 0.01)	✓ Liraglutide and dulaglutide appear to improve erectile function in men with T2D, regardless of baseline hypogonadism status ✓ Additional controlled studies are required to confirm these preliminary findings
La Vignera et al., 2023 [135]	✓ Prospective interventional comparative study ✓ Objective: To investigate the effects of liraglutide on reproductive and sexual function in men of childbearing age with metabolic hypogonadism ✓ Population: 110 men aged 18–35 with metabolic hypogonadism, divided into three groups based on fertility intentions ✓ Groups: ▪ Group A (*n* = 35): Desiring fatherhood, treated with gonadotropins (urofollitropin 150 IU three times a week, plus hCG 2000 IU twice a week) ▪ Group B (*n* = 35): not seeking fatherhood, treated with liraglutide 3 mg/day ▪ Group C (*n* = 40): already fathered a child, treated with TRT 60 mg/day ✓ Duration: 4 months	✓ Group B (liraglutide) showed: ▪ Significant improvement in conventional sperm parameters compared with baseline and Group A ▪ Improved erectile function compared with baseline and both Groups A and C ▪ Significantly ↑ levels of TT and SHBG both relative to baseline and in comparison with the other two Groups: 1. TT: 192.9% (1.4 ± 0.6 vs. 4.1 ± 0.5 ng/mL, *p* < 0.05) 2. SHBG: 157.1% (14.0 ± 3.0 vs. 36.0 ± 4.0 nmol/L, *p* < 0.05) ▪ Significantly ↑ gonadotropin levels compared with both Groups A and C: 1. FSH: Group A: 0.9 ± 0.2 IU/L vs. Group B: 2.6 ± 0.2 IU/L vs. Group C: 0.2 ± 0.1 IU/L, *p* < 0.05 2. LH: Group A: 1.0 ± 0.3 IU/L vs. Group B: 3.2 ± 0.2 IU/L vs. Group C: 0.3 ± 0.1 IU/L, *p* < 0.05	✓ Liraglutide is a promising and safe pharmacological option for men with obesity and metabolic hypogonadism, showing advantages over traditional therapies in improving both reproductive and sexual function
Lisco et al., 2024 [106]	✓ Retrospective cohort study ✓ Objective: To assess the effects of GLP-1 RAs combined with metformin vs. metformin alone on erectile function and metabolic parameters in men with T2D ✓ Population: 108 male outpatients with T2D and ED (median age 60 years) ✓ Groups: ▪ Metformin alone (*n* = 45) ▪ GLP-1RAs plus metformin (*n* = 63) ✓ Duration: 12 months	✓ After 12 months, GLP-1RAs plus metformin improved glucose control more than metformin alone: HbA1c ↓ from 8.3 ± 0.2% to 7.0 ± 0.3% (*p* < 0.0001) vs. 7.0 ± 0.5% to 7.3 ± 0.4% (*p* = 0.0007) ✓ GLP-1RAs plus metformin vs. metformin alone: ▪ Significant weight loss: −5.82 ± 0.69 kg (*p* < 0.0001) ▪ ↓ in WC: −4.99 ± 0.6 cm (*p* < 0.0001) ▪ Improvement in HbA1c: −0.56% ± 0.13% (*p* < 0.0001) ▪ ↓ in FPG: −25.54 ± 3.09 mg/dL (*p* < 0.0001) ▪ ↑ in TT: +41.41 ± 6.11 ng/dL (*p* < 0.0001) ▪ ↑ in fT: +0.44 ± 0.09 ng/dL (*p* < 0.0001) ▪ Improvement in self-reported erectile function: IIEF5 score +2.26 ± 0.26 (*p* < 0.0001) ✓ Predictors of greater IIEF5 improvement: ▪ Higher baseline IIEF-5 score (*p* = 0.045) ▪ Presence of carotid stenosis (*p* = 0.045) ▪ Greater weight loss from baseline (*p* = 0.013) ✓ 1-year GLP-1RAs plus metformin treatment was the main determinant of improved IIEF5 score (2.74 ± 0.53, *p* < 0.0001)	✓ GLP-1RAs combined with metformin improved erectile function in men with T2D compared with metformin alone, independent of baseline characteristics ✓ Improvements may be due to positive vascular effects ✓ Due to the retrospective nature of the study, a definitive cause-effect relationship cannot be established; RCTs are needed to confirm the efficacy of GLP-1RAs in treating ED in T2D
Able et al., 2025 [140]	✓ Retrospective cohort study ✓ Objective: To evaluate the risk of newly diagnosed ED or PDE5 inhibitor use, and testosterone deficiency, following at least one month of semaglutide prescription in non-diabetic men using it for weight loss ✓ Population: Population: 3094 non-diabetic men with obesity, aged 18–50, who received semaglutide, matched 1:1 to non-diabetic men with obesity without semaglutide exposure	✓ ED and/or PDE5 inhibitor use: ↑ in semaglutide users vs. non-users (1.47% vs. 0.32%; RR 4.5, 95% CI 2.3–9.0) ✓ Testosterone deficiency: ↑ in semaglutide users vs. non-users (1.53% vs. 0.80%; RR 1.9, 95% CI 1.2–3.1)	✓ ED, PDE5 inhibitor prescription, and testosterone deficiency were more frequent in non-diabetic men with obesity on semaglutide, despite weight loss benefits ✓ Patients should be informed of potential sexual health risks, particularly those at higher risk ✓ Further research is needed to clarify underlying mechanisms and optimize treatment outcomes-
Pourabhari Langroudi et al., 2025 [107]	✓ Pharmacovigilance study using the FAERS database (2003–2024) ✓ Objective: To investigate the association between GLP-1 RAs and male sexual dysfunction, including: ▪ ED ▪ Orgasmic dysfunction ▪ ↓ libido	✓ Total cases: 182 ✓ Most common indication: T2D (43.9%), followed by weight loss (4.9%) ✓ Patient age: Predominantly 40–60 years ✓ Trends over time: Reports ↑ in recent years, especially 2023; semaglutide reports rising, tirzepatide surpassed semaglutide in 2023 and Q1 2024 ✓ Subcutaneous route of administration most common ✓ Drug distribution: Exenatide 24.2%, lixisenatide 0.5%, liraglutide 20.9%, dulaglutide 18.7%, semaglutide 21.4%, tirzepatide 14.3% ✓ Type of sexual dysfunction: ED: 71.4%; ↓ libido: 15.1%; orgasmic abnormalities: 1.6%; other sexual dysfunctions: 11.0%	✓ The link between GLP-1 RAs and male sexual dysfunction appears weak, with current evidence not indicating a significant clinical risk ✓ Nevertheless, ongoing monitoring is advised as GLP-1 use grows, while further research is needed to clarify their impact on sexual function

Abbreviations: BMI: body mass index; CI: 95% confidence interval; CVD: cardiovascular disease; ED: erectile dysfunction; eGFR: estimated glomerular filtration rate; FAERS: FDA Adverse Event Reporting System; FPG: fasting plasma glucose; FSH: follicle-stimulating hormone; fT: free testosterone; GLP-1 RA: glucagon-like peptide-1 receptor agonist; HbA1c: glycated hemoglobin A1c; HPT: hypothalamic–pituitary–testicular; HR: hazard ratio; hCG: human chorionic gonadotropin; IIEF: international index of erectile function; LH: luteinizing hormone; LS: least squares; MBS: metabolic bariatric surgery; MetS: metabolic syndrome; PDE5: phosphodiesterase type 5; Q1: first quartile; RCT: randomized controlled trial; REWIND: Researching Cardiovascular Events With a Weekly INcretin in Diabetes; RR: relative risk; SHBG: sex hormone-binding globulin; T2D: type 2 diabetes; TRT: testosterone replacement therapy; TT: total testosterone; WC: waist circumference; ↑: increase; ↓: decrease.

**Table 2 biomolecules-15-01284-t002:** Evidence from recently published meta-analyses on the comparative role of GLP-1 receptor agonists in erectile dysfunction relative to other antidiabetic therapies.

Author, Year, Reference	Yang et al., 2025 [171]	Rastrelli, et al., 2025 [172]	Corona, et al., 2025 [173]	Salvio et al., 2025 [175]
**Meta-analysis characteristics**	✓ Objective: To evaluate the impact of different classes of antidiabetic drugs on ED in patients with diabetes ✓ 3 cohort studies included ✓ Number of participants: 314 (209 eligible) men ✓ Baseline characteristics: ▪ Mean age: 58.9 years ▪ Mean BMI 32.1 kg/m^2^ ✓ Mean follow-up: 2.3 years	✓ Objective: To assess the impact of antidiabetic drugs on ED in individuals with diabetes or prediabetes ✓ 3 observational and 3 placebo-controlled RCTs included ✓ Number of participants: 4112 subjects ✓ Baseline characteristics: ▪ Mean age: 57.8 years ▪ Mean BMI: 31.4 kg/m^2^ ▪ Mean testosterone: 13.1 nmol/L ✓ Mean follow-up: 87.4 weeks	✓ Objective: To explore the effects of GLP-1 RAS on the HPT axis and male sexual function ✓ 6 studies included (type not defined) ✓ Number of participants: 386 individuals ✓ Baseline characteristics: ▪ Mean age 47.1 ± 11.1 years ▪ Mean BMI 35.2 ± 3.9 kg/m^2^ ✓ Mean follow-up: 31.4 ± 19.1 weeks	✓ Objective: To investigate the effects of GLP-1 RAs on hormone secretion in overweight and men with obesity, and to compare their impact on testicular and erectile function with other antidiabetic or weight-loss therapies ✓ 6 cohorts and 1 randomized open-label study included ✓ Number of participants: 680 overweight/men with obesity ✓ Baseline characteristics: ▪ Mean age: 47.65 years ▪ Mean BMI: 34.44 kg/m^2^ ✓ Mean follow-up: 8.43 months
**Results**	✓ GLP-1RAs were more effective than metformin in improving ED in diabetic patients (*p* = 0.02) and the effect was more pronounced in patients with higher BMI (*p* = 0.02) ✓ TZDs may offer benefits in sexual function, though further safety and efficacy studies are needed ✓ Insulin and SGLT-2is showed potential benefit but lack strong supporting evidence ✓ The effect of metformin and SUs remains unclear due to mixed or inconclusive data	✓ Observational studies: ▪ Metformin, GLP1 RAs, and SGLT2is were associated with improved ED ▪ No significant benefit compared to diet (for all drugs including SUs and DPP-4is) ✓ RCTs showed significant improvement in IIEF score with metformin, pioglitazone, and dulaglutide	✓ GLP-1 RA use was associated with: ▪ Significant ↑ in TT, calculated fT, and SHBG ▪ ↑ gonadotropins (either LH or FSH) at the study endpoint ▪ Improved erectile function across studies ✓ Meta-regression analysis: ▪ Significant positive correlation between the degree of GLP-1RA–induced weight loss and testosterone ↑ (*p* < 0.0001) ▪ No significant heterogeneity (*p* = 0.429)	✓ GLP-1RAs significantly ↑ TT levels (*p* < 0.0001), with similar ↑ in FT, SHBG, LH, and FSH ✓ Weight, BMI, WC, and HbA1c were significantly ↓ following GLP-1RA treatment ✓ Meta-regression analysis showed a negative correlation between TT ↑ and ↓ in weight and BMI ✓ Compared with other treatments, GLP-1RAs had similar effects on serum androgens, but greater ↓ in BMI and greater ↑ in gonadotropins and erectile function indices
**Conclusions**	✓ GLP-1 RAs appear to outperform metformin in improving erectile function in diabetic men, especially those with obesity ✓ Other drug classes may also offer positive effects on ED, but more high-quality clinical trials are needed to confirm these findings and address safety considerations	✓ Given the link between ED and future CVD risk, GLP-1 RAs and SGLT-2is are considered the preferred options, due to their broad role in reducing long-term complications ✓ Metformin is a suitable alternative for less complex patients	✓ GLP-1 RAs may offer a viable alternative therapeutic option for men with severe obesity-related testosterone deficiency, showing both hormonal and sexual function benefits alongside weight loss	✓ GLP-1RAs may play a beneficial role in managing functional hypogonadism and improving erectile function in overweight and men with obesity, primarily through weight loss–related mechanisms ✓ Current evidence does not confirm a direct effect of GLP-1 RAs on testicular function due to limitations in study design and available data

Abbreviations: BMI: body mass index; CVD: cardiovascular disease; DPP-4i; dipeptidyl peptidase-4 inhibitor; ED: erectile dysfunction; fT: free testosterone; FSH: follicle-stimulating hormone; GLP-1 RA: glucagon-like peptide-1 receptor agonist; HbA1c: glycated hemoglobin A1c; HPT: hypothalamic–pituitary–testicular; IIEF: international index of erectile function; LH: luteinizing hormone; RCT: randomized controlled trial; SHBG: sex hormone-binding globulin; SGLT-2i: sodium–glucose cotransporter-2 inhibitor; SU: sulfonylurea; TT: total testosterone; TZD: thiazolidinedione; WC: waist circumference; ↑: increase; ↓: decrease.

## Data Availability

Not applicable.

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
