# Peer review of "The Impact of Glucagon-like Peptide-1 Receptor Agonists on Erectile Function: Friend or Foe?"

_biomolecules, 2025, doi:10.3390/biom15091284_

Round 1
Reviewer 1 Report
Comments and Suggestions for Authors
This is a beautifully written paper. I have only a couple of comments.
A drop of 192.9% in TT and a rise of 157.1% in SHBG might only result in a significant rise in free T in a minority of patients achieving the greatest weight loss. Therefore patients patients with very low T (say 4nmol/l) pre-treatment may not see significant rises in FT.
Do the authors suggest measuring TT and FT prior to treatment to detect these patients , and treat with TTh if necessary?
FT is the best predictor for ED and CV risk according to EMAS.
The impact on lean muscle mass is not fully discussed and may turn out o be the major drawback. Lean muscle loss has been shown to be associated with ED, so perhaps, patients with greatest loss of lean muscle might explain the paradox in ED response, but published trials might not have detected this.
Author Response
REVIEWER 1
This is a beautifully written paper. I have only a couple of comments.
Answer:
We sincerely thank the reviewer for the kind words and the insightful comments. The manuscript has undergone substantial revisions at multiple levels (accuracy of data, table and figure data, and use of the English language) to enhance its clarity and precision. Our detailed responses are as follows:
- A drop of 192.9% in TT and a rise of 157.1% in SHBG might only result in a significant rise in free T in a minority of patients achieving the greatest weight loss. Therefore patients with very low T (say 4nmol/l) pre-treatment may not see significant rises in FT. Do the authors suggest measuring TT and FT prior to treatment to detect these patients ,and treat with TTh if necessary? FT is the best predictor for ED and CV risk according to EMAS.
Answer:
As internists/diabetologists, we are not specialists in the specific field addressed by this question. However, the reviewer’s remark was particularly valuable, as it prompted us to provide a more comprehensive recommendation for the management of these patients, clarifying the roles of the urologist and the diabetologist. In this context, we have briefly incorporated into our manuscript the most recent guidelines of the European Association of Urology (2025) (lines 792–813). Additionally, the previous part of the manuscript referring to testosterone replacement therapy has been removed (pathophysiology section).
- The impact on lean muscle mass is not fully discussed and may turn out to be the major drawback. Lean muscle loss has been shown to be associated with ED, so perhaps, patients with greatest loss of lean muscle might explain the paradox in ED response, but published trials might not have detected this.
Answer:
We also acknowledge the second comment as highly important and insightful. It was indeed an omission on our part not to include this information earlier. We have now addressed it appropriately in the revised version (lines 776–791).
Reviewer 2 Report
Comments and Suggestions for Authors
This is a comprehensive, well-structured narrative review that synthesizes current evidence on GLP-1 RAs and erectile dysfunction (ED) in diabetes. It balances mechanistic insights with clinical data, addressing conflicting findings while identifying research gaps. The paper is timely given the expanding use of GLP-1 RAs for diabetes/obesity.
Major concerns:
- This review relies on small, short-term, observational studies. Only one RCT (REWIND) assessed ED as an exploratory endpoint.
- No clear explanation for adverse effects.
- Redundancy in discussing weight loss mechanisms (Sections 4.1, 5, 7.2).
- There is lack of discussion of ED in type 1 diabetes or non-obese individuals.
Author Response
REVIEWER 2
This is a comprehensive, well-structured narrative review that synthesizes current evidence on GLP-1 RAs and erectile dysfunction (ED) in diabetes. It balances mechanistic insights with clinical data, addressing conflicting findings while identifying research gaps. The paper is timely given the expanding use of GLP-1 RAs for diabetes/obesity.
Answer:
We sincerely thank the reviewer for the kind words and the insightful comments. The manuscript has undergone substantial revisions at multiple levels (accuracy of data, table and figure data, and use of the English language) to enhance its clarity and precision. Our detailed responses are as follows:
Major concerns:
- This review relies on small, short-term, observational studies. Only one RCT (REWIND) assessed ED as an exploratory endpoint.
Answer:
Indeed, this constitutes the main limitation of the present study, which we have already highlighted at several points in the manuscript. However, unlike other articles in the literature that emphasize the beneficial role of GLP-1 RAs in ED, we aimed to take a more comprehensive approach to the issue, allowing for more reliable conclusions based on the existing data and providing guidance for future research. In this context, the conclusion section (including the final accompanying figure) of our manuscript (lines 844–855) has undergone minor revisions to make the message clearer.
- No clear explanation for adverse effects.
Answer:
The section on adverse effects has been expanded to include an interpretation of adverse effects (lines 349-392).
- Redundancy in discussing weight loss mechanisms (Sections 4.1, 5, 7.2).
Answer:
- In Sections 4.1 and 7.2, redundancy has been addressed.
- In Section 5, the majority of the information concerns the vascular effects of GLP-1 RAs. Therefore, we believe that removing the limited available data on weight loss would not benefit our manuscript, given that body weight represents an important factor in the development of ED.
- There is lack of discussion of ED in type 1 diabetes or non-obese individuals.
Answer:
This manuscript discusses the effect of GLP-1 RAs on erectile function. Given that the approved indications of these agents are for T2D and obesity, and not for T1D or non-obese individuals, the pathophysiology section of ED focuses primarily on these populations, as it provides readers with the necessary background to follow the rest of the text. Nevertheless, irrespective of diabetes type, chronic hyperglycemia is a key mechanism underlying ED (highlighted more clearly in lines 164–165). Furthermore, the potential for metabolic syndrome/obesity within the context of ‘’double diabetes’’ in patients with T1D has already been noted (lines 244–245).
Reviewer 3 Report
Comments and Suggestions for Authors
There are few reviews that comprehensively summarize the impact of glucagon-like peptide-1 receptor agonists (GLP-1 RAs) on erectile dysfunction (ED). This paper is valuable in that it addresses both basic and clinical research, while also maintaining balance by including reports of potential adverse effects, which adds to its originality. The structure flows clearly from background to mechanisms, basic research, clinical studies, comparisons with other drugs, and conclusions, with well-organized figures and tables that aid visual understanding. It cites a wide range of evidence, including major RCTs, observational studies, and pharmacovigilance data, although direct data on the relationship between semaglutide or tirzepatide and ED remain scarce, which is a limitation. While it discusses both beneficial and harmful effects, the critique of each study’s limitations and potential biases is somewhat superficial, and clearer caution is warranted regarding the weakness of causal inference in observational studies and FAERS data. Overall, this is a useful review for clinicians and could help inform future research design and hypothesis generation, but the conclusions should be stated cautiously, and at present, the stance of “promising but not yet conclusive” is appropriate.
Author Response
REVIEWER 3
There are few reviews that comprehensively summarize the impact of glucagon-like peptide-1 receptor agonists (GLP-1 RAs) on erectile dysfunction (ED). This paper is valuable in that it addresses both basic and clinical research, while also maintaining balance by including reports of potential adverse effects, which adds to its originality. The structure flows clearly from background to mechanisms, basic research, clinical studies, comparisons with other drugs, and conclusions, with well-organized figures and tables that aid visual understanding. It cites a wide range of evidence, including major RCTs, observational studies, and pharmacovigilance data, although direct data on the relationship between semaglutide or tirzepatide and ED remain scarce, which is a limitation. While it discusses both beneficial and harmful effects, the critique of each study’s limitations and potential biases is somewhat superficial, and clearer caution is warranted regarding the weakness of causal inference in observational studies and FAERS data. Overall, this is a useful review for clinicians and could help inform future research design and hypothesis generation, but the conclusions should be stated cautiously, and at present, the stance of “promising but not yet conclusive” is appropriate.
Answer:
We sincerely thank the reviewer for the kind words and the insightful comments. The manuscript has undergone substantial revisions at multiple levels (accuracy of data, table and figure data, and use of the English language) to enhance its clarity and precision. Sections 6 (lines 503-595) and 8 (lines 707-857) have been revised in order to provide a more accurate presentation of both the results and the limitations of the available studies.